# Wild Bird Surveillance in the Gauteng Province of South Africa during the High-Risk Period for Highly Pathogenic Avian Influenza Virus Introduction

**DOI:** 10.3390/v14092027

**Published:** 2022-09-13

**Authors:** Celia Abolnik, Thandeka P. Phiri, Gerbrand van der Zel, Jade Anthony, Nadine Daniell, Liesl de Boni

**Affiliations:** 1Department of Production Animal Studies, Faculty of Veterinary Science, University of Pretoria, Onderstepoort 0110, South Africa; 2Gauteng Department of Agriculture and Rural Development, Johannesburg 2000, South Africa

**Keywords:** wild bird surveillance, avian influenza, H9N2, H5N1, environmental faecal sampling

## Abstract

Migratory birds carried clade 2.3.4.4B H5Nx highly pathogenic avian influenza (HPAI) viruses to South Africa in 2017, 2018 and 2021, where the Gauteng Province is a high-risk zone for virus introduction. Here, we combined environmental faecal sampling with sensitive rRT-PCR methods and direct Ion Torrent sequencing to survey wild populations between February and May 2022. An overall IAV incidence of 42.92% (100/231) in water bird faecal swab pools or swabs from moribund or dead European White Storks (*Ciconia ciconia*) was detected. In total, 7% of the IAV-positive pools tested H5-positive, with clade 2.3.4.4B H5N1 HPAI confirmed in the storks; 10% of the IAV-positive samples were identified as H9N2, and five complete H9N2 genomes were phylogenetically closely related to a local 2021 wild duck H9N2 virus, recent Eurasian LPAI viruses or those detected in commercial ostriches in the Western and Eastern Cape Provinces since 2018. H3N1, H4N2, H5N2 and H8Nx subtypes were also identified. Targeted surveillance of wild birds using environmental faecal sampling can thus be effectively applied under sub-Saharan African conditions, but region-specific studies should first be used to identify peak prevalence times which, in southern Africa, is linked to the peak rainfall period, when ducks are reproductively active.

## 1. Introduction

Avian influenza is one of the largest threats to poultry production globally, and food security in developing countries. Outbreaks on poultry farms lead to mass depopulations and international trade restrictions of live birds and poultry products, causing severe economic losses [1]. Moreover, some of the influenza A viruses (IAVs) that cause avian influenza are zoonotic with high lethality, and predictions are that the next human pandemic may be caused by an avian influenza virus [2]. Wild birds of the *Anseriformes* (ducks, geese and swans) and, to a lesser extent, *Charadriiformes* (gulls and shorebirds) orders are the primordial hosts and reservoir for IAVs in natural ecosystems [3]. IAV is classified within the *Alphainfluenza* genus of the *Orthomyxoviridae* family, possessing an eight-segmented, negative sense RNA genome [4]. Sixteen haemagglutinin (H) and nine neuraminidase (N) virus subtypes are described in avian species, coexisting in populations as genetically diverse subtypes in their natural low pathogenic form (LPAI) that manifests as sub-clinical infections. IAV diversity in wild birds is driven by a dynamic evolutionary process of genomic reassortment and genetic drift [3,5]. The viruses replicate in the hosts’ respiratory and gastrointestinal tracts, from where they are excreted into the environment, and if transmitted to terrestrial poultry species, the H5 and H7 LPAI subtypes can mutate to highly pathogenic avian influenza (HPAI) viruses, a highly contagious and lethal form that replicates systemically to cause high morbidity and mortality in susceptible species [3,6]. In 1996, a strain of H5N1 HPAI, termed the Goose/Guangdong (Gs/GD) virus emerged in Chinese poultry and has since evolved into nine clades and multiple sub-clades. In 2014, in an unprecedented event, Gs/GD lineage H5 HPAI viruses adapted to and became established in wild birds, subsequently spreading globally along bird migratory flyways, causing catastrophic multiple pandemic waves across Asia, Europe, Africa and North America since 2016. Gs/GD clade 2.3.4.4 emerged as a particularly robust viral lineage with a wide geographical distribution [7,8,9].

South Africa, situated at the southern tip of the African continent, acts as an ecological sink for LPAI viruses of Eurasian origin [10]. IAVs are introduced into the equatorial regions of Africa by overwintering Palaearctic ducks or shorebirds and are subsequently spread intra-continentally by Afro-tropical duck species [1,11]. Thus, once the clade 2.3.4.4 Gs/GD H5 HPAI viruses entered the Eurasian migratory flyways, southern Africa could not escape the HPAI pandemic. The climate-driven migrations of Afro-tropical duck vectors introduced clade 2.3.4.4B HPAI strains into the north-central regions of South Africa in 2017 (H5N8), 2018 (H5N8) and 2021 (H5N1) [12,13,14,15]. Once the local wild bird populations became infected, the viruses disseminated to other provinces within weeks, and devastating outbreaks followed in chickens of the layer and broiler industries, commercial ostriches and subsistence poultry with economic losses totalling hundreds of millions of US dollars. The outbreaks in 2018 and 2021 also exacted a heavy ecological toll on susceptible wild coastal bird populations, of which some species are critically endangered [12,14,15].

The incidence of clade 2.3.4.4 H5Nx viruses in wild birds and poultry in the Eurasian feeder zone for South Africa has not decreased with time, instead the number of wild bird and poultry outbreaks in Europe seem to be increasing annually with concerns raised for an enzootic status of HPAIV in northern Europe [9]. Since the start of the HPAI pandemic, the World Organization for Animal Health (WOAH) has strongly encouraged its member states to intensify their active wild bird surveillance activities as an early warning system and to track the emergence of new and more dangerous HPAI strains [6,8]. Yet, despite the high risk of annual introductions of HPAI viruses from the northern hemisphere, South Africa still lacks a well-resourced, coordinated national wild bird surveillance program, with this responsibility left to provincial veterinary authorities, the poultry industry and academic institutions.

The Gauteng Province, situated north of the Vaal River in the Highveld region, is the smallest of South Africa’s nine provinces, covering just 1.5% of the country’s land area, but it is the most urbanized and also houses 10.1% of South Africa’s broilers and 28.5% of its layers in the highest density of layer farms and layer hens in the country (Figure 1; [16]). The index cases in the 2017, 2018 and 2021 H5Nx HPAI outbreaks were within or very near the borders of the Gauteng province [13,14,15], which suggests that the climatic zone in which Gauteng falls may be an important destination or stopover for the Afro-tropical ducks arriving from East and Central Africa. Gauteng is therefore considered a high-risk zone and potential hotspot for the introduction and transmission of HPAI viruses [15], making it a key surveillance point in the country. This province has a rich biodiversity of wild birds, with 496 species recorded including high counts of target Afro-tropical Anseriform species such as the Cape Teal (*Anas capensis*), Egyptian Goose (*Alopochen aegyptiacus*), African Black duck (*Anas sparsa*), South African Shelduck (*Tadorna cana*), Red-billed Teal (*Anas erythrorhyncha*), Yellow-billed Duck (*Anas undulata*), Knob-billed Duck (*Sarkidiornis melanotos*), White-faced Whistling Duck (*Dendrocygna viduata*) and African Pygmy Goose (*Nettapus auritus*) [17]. IAVs of various subtypes were detected in some of these species and in African Sacred ibises (*Threskiornis aethiopicus*) in South Africa during active and passive surveillance studies [10,14,18,19,20].

IAV dynamics are driven by the presence of immunologically naïve juvenile birds in the population that shed the highest quantities of virus, the density of host congregations, interspecies mixing and environmental conditions (especially temperature) that affect the survival of virus outside of its host [11]. Active wild bird surveillance should therefore be targeted towards times of year, species and locations in which infection is more likely [6], but the seasonality of IAV prevalence in the hosts in Africa is much less pronounced than in the northern hemisphere’s temperate and boreal regions, with the consequent possibility for viral transmission throughout the year [11,21]. This is because the movements, congregation and the timing of breeding of most Afro-tropical waterfowl species are driven by rainfall patterns instead of temperature [11,22,23], and their extended and asynchronous breeding seasons produce a continuous introduction of juvenile birds into the community in contrast to the more synchronized breeding periods of species in the northern hemisphere that lead to a seasonal pulse of juveniles [11,24].

Prior longitudinal surveillance studies in the southern African region failed to detect specific seasonal peaks in the prevalence of IAV in Afro-tropical birds [11,21,25], but the study sites in Botswana, South Africa, Mozambique, Zambia and Zimbabwe were separated over thousands of kilometres and a range of climatic zones, and the numbers of samples per site that could be collected were limited by the invasive methods of trapping and swabbing birds, in often remote locations. Alternatively, it is possible that seasonal IAV peaks in southern Africa are region-specific and would be revealed by testing larger numbers of samples from specific climatic zones. Thus, environmental faecal sampling, used successfully in surveillance initiatives for IAVs across the world since the 1970′s [26], was applied at the University of Pretoria in 2016–2017 in a longitudinal study of a wild population of Egyptian Geese at a site in Pretoria, Gauteng Province. It revealed a high incidence (>80%) of IAV in this population in the summer months of January and February, coincidentally the highest rainfall period in the Highveld, and a small peak (0.3%) in July (winter) [19]. The high incidence of IAV in January and February was confirmed with follow-up sampling at the site in 2021, when 85% and 25% IAV prevalence, respectively, was detected in the same population [20].

In view of this new information and considering that the index cases for the H5Nx HPAI outbreaks were recorded in mid-February (2018), mid-April (2021) and early June (2017) [13,14,27], the aim of the present study was to conduct wild bird surveillance in the Gauteng Province from February to May 2022, the peak prevalence and high-risk period for introduction of new IAVs. Our sampling strategy was based on non-invasive environmental faecal sampling, targeting sites where heterogeneous populations of wild water birds congregate. We identified and/or isolated the IAV subtypes present and used deep sequencing to obtain whole genomes for phylogenetic comparisons.

## 2. Materials and Methods

### 2.1. Sample Collection Sites and Study Period

Permanent water bodies in reserves, public recreational areas, a municipal dump, or private property in the Gauteng Province were targeted where the regular presence of a large number of wild waterfowl, preferably heterogeneous species, was observed. Accessibility to fresh faecal droppings on banks or under trees used for overnight roosting was an important consideration. Ultimately thirteen sites were selected, four around the City of Tshwane (Pretoria), seven in and around the city of Johannesburg and two on the southern provincial border (Figure 1). Sampling was conducted from February to May 2022. One site each in Johannesburg (Zoo Lake) and Pretoria (Bon Accord Dam) was sampled weekly in February and March, whereas the others were typically sampled once a month. The study was conducted under the approval of the University of Pretoria Research and Animal Ethics Committees (project no. REC163-21) and the Department of Agriculture and Rural Development (Section 20 permit no. 12/11/1/1/-2131SS) with site access authorized by the relevant owner or authority.

### 2.2. Sample Collection

Sample collection from fresh faeces was performed in the mornings to avoid prolonged environmental exposure. Individual droppings were sampled with sterile rayon-tipped swabs (Copan, Brescia, Italy), pooling five swabs into 13 mL sterile polypropylene tubes (Carl Roth GmbH, Karlsruhe, Germany) pre-filled with 3 mL of viral transport medium (VTM) (brain-heart infusion broth (pH 7.2) (Oxoid, ThermoFisher Scientific, Waltham, USA), 10% (*v*/*v*) glycerol 100 mg/L doxycycline (Mylan), 100 mg/L enrofloxacin (Cipla), 1000 mg/L penicillin-streptomycin (Sigma-Aldrich), 5 mg/L Amphotericin B (Bristol-Myers Squibb)). Alternatively, swabs were pooled into commercial 3 mL VTM Tubes for Viruses, Chlamydia, Mycoplasma and Ureaplasma (Copan, Brescia, Italy). Two to ten swab pools were collected per site and transported in a cooler on ice packs to the University of Pretoria and kept at 4 °C degrees until testing within 24 to 72 h.

### 2.3. Nucleic Acid Extractions and Real Time RT-PCRs

Swab pools were thoroughly vortexed prior to total nucleic acid extraction from 200 μL of the fluid using either IndiMag Pathogen Kits in an IndiMag™ 48 instrument (Indical BioSciences, Leipzig, Germany) or TriZOL reagent (Invitrogen, ThermoFisher Scientific, Waltham, USA), according to the manufacturers’ recommended methods. Total nucleic acid or RNA was eluted or resuspended, respectively, in 0.1 mL volumes of elution buffer. Nucleic acid extracts were screened by real time RT-PCR (rRT-PCR) for the presence of the IAV group using VetMax™-Gold AIV Detection Kits (ThermoFisher Scientific, Waltham, USA) in a StepOnePlus™ instrument (Life Technologies, Thermo Fisher Scientific, Waltham, USA) according to the recommended procedure, but using half volume reaction mixes with 4 μL of test or control RNA, and nuclease-free water as the no-template control. RT-PCR cycle threshold (Ct) values < 40 were considered positive. The limit of detection of the VetMax™-Gold AIV RT-qPCR assay is 10 ^0.8^ EID_50_ units/mL [28].

Samples that tested IAV positive were subsequently tested with rRT-PCR assays for the H5 and H9 subtypes, with VetMAX Plus RT-PCR kits (ThermoFisher Scientific) according to the manufacturer’s recommended protocol and a 53 °C annealing temperature. For H9 subtype detection, the oligonucleotide primers and FAM-labelled probe described by Hoffmann et al. [29] were used, and for H5 subtype detection the following primer and probe set was used (all sequences in the 5′ to 3′ orientation): H5 Forward: ACR TAT GAC TAC CCW CAR TAT TCA G; H5 Reverse: AGA CCA GCY AYC ATG ATT GC; H5 probe: FAM– TCW ACA GYG GCG AGT TCC CTA GCA –MGB. The underlined bases are modifications made to the original validated pair [30], to increase the sensitivity in detecting clade 2.3.4.4b H5Nx viruses circulating in wild birds (multiple sequence alignments not shown).

Where samples were identified as H5-positive by real-time RT-PCR, the conventional H5-specific RT-PCR with the oligonucleotide primer pair J3/B2a [30] was performed in a 20 μL volume with 10 X Phusion Flash High Fidelity PCR Master mix (Thermo Fisher Scientific), 60 U MMLV reverse transcriptase (Invitrogen) and 6 U Human Placental Ribonuclease Inhibitor (ThermoFisher Scientific). The RT-PCR products were separated in 2% agarose gels stained with ethidium bromide. The 300 bp product was excised and purified using a QiaQuick gel extraction kit (Qiagen, Aarhus, Denmark) and submitted to Inqaba Biotech (Pty) Ltd., Pretoria, South Africa for Sanger DNA sequencing. To identify the other subtypes, viz. H1-H4, H6-H8, H10-H16 and N1-N9, Superscript III Platinum SYBR Green One-Step qRT-PCR Kits with ROX (Invitrogen, Thermo Fisher Scientific) were used according to the manufacturer’s recommended procedure with the H and N subtype primer sets and conditions described by Hoffmann et al. [29]. Melt curve analysis was performed to compare the test sample with the appropriate subtype positive controls and no-template negative controls. Only samples with IAV group Ct values ≤35 were tested with SYBR-Green RT-PCRs.

### 2.4. Virus Isolation

Sample pools with Ct values ≤30 [26,31] were either inoculated into 9-to-11-day-old embryonated Specific Pathogen Free (SPF) hen’s eggs (AviFarms, Pretoria, South Africa) for virus isolation according to the WOAH-recommended method (2019), or in a confluent QH9-2/1 Quail cells (Nuvonis, Vienna, Austria), according to the supplier’s recommended procedures.

### 2.5. Genome Sequencing and Phylogenetic Analysis

Complete IAV genomes were amplified in single tube RT-PCR reactions from extracted RNA as per the method described by Zhou et al. [32], analysed by 1% agarose gel electrophoresis and submitted to the University of Stellenbosch Central Analytical Facility for Ion Torrent sequencing as described previously [14]. Ion Torrent reads were imported into the CLC Genomics Workbench 5.2.1 (QIAGEN CLC bio, Aarhus, Denmark; http://www.clcbio.com, accessed on 1 July 2022) and assembled to reference sequences representing IAV subtypes H1 to H16, N1 to N9 and the six internal gene segments (Genbank accession numbers KY621531-KY621538, JN696316, KF313565, MH412114, GU122032, MH637350, KM054845, MH637361, HE802739, KT777901, KP087869, HE802715, KX101133, MH637340, KP287772, MH637353, KM244048, MH411954, CY080155, KM244094, MH637406, KM244102, MH637420 and KJ484622).

For phylogenetic reconstruction, reference sequences were retrieved by BLAST of the NCBI (https://www.ncbi.nlm.nih.gov/nucore, accessed on 1 July 2022) and GISAID Epiflu databases and aligned in MAFFT v.7 (https://mafft.cbrc.jp/alignment/server/, accessed on 1 July 2022). Maximum likelihood (ML) phylogenetic trees were generated with the Tamura-Nei model in MEGA-X (v. 10.2.5) [33], with 1000 bootstrap replicates. Trees were drawn to scale, with branch lengths measured in the number of substitutions per site. The time to most recent common ancestor (tMRCA) was determined from dated maximum clade credibility (MCC) trees produced for each of the eight genome segments in BEAST v.2 software [34]. MCC trees were reconstructed using a Hasegawa–Kishono–Yano (HKY) nucleotide substitution model with a gamma distribution of substitution rates, a Coalescent Bayesian Skyline model and a Relaxed Lognormal clock. Markov chain Monte Carlo chains of between 50 and 80 million iterations were performed and assessed with Tracer v1.7.2 [35] to ensure that an effective sample size (ESS) of >200 was achieved, with statistical uncertainty of the nodes reflected in values of the 95% highest posterior density (HPD). MCC trees with common ancestor heights were summarized using TreeAnnotator v.2.6.6 and visualized using FigTree v.1.4.2 (http://tree.bio.ac.uk/software/figtree/, accessed on 15 July 2022).

## 3. Results

### 3.1. Detection and Identification of IAV in Wild Bird Faecal Samples

Two hundred and thirty-one (231) faecal swab pools comprising 1155 individual samples were collected and tested between February and May 2022, plus the pooled oropharyngeal and cloacal swabs from one moribund European White Stork (*Ciconia ciconia*) found at Bon Accord Dam, Pretoria on 8 February 2022, and a European White Stork fresh carcass found at Turffontein, Johannesburg on 23 February 2022. About ten European White Stork carcasses in varying states of decomposition were observed at the latter site, with evidence of scavenging on some, so it appeared that mortalities had been ongoing for some time. Of the total swab pools, 100/233 (42.92%) tested positive for the presence of IAV-specific RNA, with an average Ct value of 35.7 (range 18.53–39.57) (Appendix A). IAV was detected at all study sites apart from the Heidelberg Dump site, but this site was only sampled twice with three pools collected.

Of the 100 IAV-positive swab pools, seven (7%) tested positive for the H5 subtype using rRT-PCR, including the European White Storks at Bon Accord Dam and Turffontein. A full genome was successfully amplified from the swabs of the moribund bird at Bon Accord Dam (BA107, H5 Ct = 17.74) and the complete genome was assembled from Ion Torrent sequencing reads, with deep coverage. This virus was successfully isolated in embryonated chicken eggs after two passages. A/South Africa/European White Stork/BA107/2022 was identified as a clade 2.3.4.4B H5N1 HPAI virus by BLAST analysis, but the phylogenetic relationships of this viruses with other clade 2.3.4.4B H5N1 strains from the 2021/2022 outbreaks in South Africa will be described elsewhere. Full genome amplification and sequencing was unsuccessful for the stork carcass at Turffontein (TRF001, H5 Ct = 33.34), but a partial H5 sequence was obtained by conventional RT-PCR, with an HA_0_ cleavage site motif of PLREKRRKRGLF, confirming the HPAI pathotype, with the neuraminidase identified as N1 with subtype-specific SYBR Green rRT-PCR. The H5N1 subtype was also identified by rRT-PCR assays at Zoo Lake on 11 March (ZL506; H5 Ct = 34.91) and Bon Accord Dam on 15 March (BA611; H5 Ct = 35.86) but attempts at amplifying the partial HA gene for pathotype determination were unsuccessful. Similarly, the pathotype of an H5N2 virus (ZL206; H5 Ct = 34.02) detected at Zoo Lake on 17 February is undetermined, and two other H5 viruses were identified for which the N-type could not be established, sampled on 22 February at Bon Accord Dam (BA303, H5 Ct = 36.95) and 19 April at Loch Vaal, Vereeniging (D5513-2; H5 Ct = 35.96).

Ten (10%) of the 100 IAV-positive samples tested positive for the H9 subtype by rRT-PCR, at three of the thirteen sites sampled in the province, with the N2 subtype identified with subtype-specific SYBR Green rRT-PCRs. The H9N2 virus was first detected in four swab pools at Zoo Lake Johannesburg on the 11th of February (ZL117, H9 Ct = 24.71; ZL118, H9 Ct = 25.17; ZL119, H9 Ct = 35.85; ZL121, H9 Ct = 24.48) and then weekly in one or two swab pools over the next four consecutive sampling expeditions until the 4th of March (ZL213, H9 Ct = 36.44; ZL309, H9 Ct = 26.01; ZL310, H9 Ct = 36.90; ZL410, H9 Ct = 36.64). Finally, H9N2 viruses were identified at two other sites, namely Loch Vaal, Vereeniging on 19 April (D5513-1, H9 Ct = 35.93) and on 9 May at Rooikraal, Heidelberg (D6539-2, H9 Ct = 37.0). Full IAV genomes were successfully amplified from the four Zoo Lake cases with low Ct values (ZL117, ZL118, ZL121 and ZL309), but surprisingly, a full genome could also be amplified from ZL213 that had a Ct value of only 36.44. Full genome amplification products could not be obtained for the other H9N2-positive samples. Complete H9N2 genome sequences, with coverage that ranged from 4472× to 482684× per genome segment, were assembled from Ion Torrent sequencing reads for the five viruses, and A/South Africa/wild duck/ZL309/2022 (H9N2) was selected and successfully isolated in QH9 quail cells.

All samples that were not identified as H5 or H9, with Ct values <37 were tested with subtype-specific primers for H1-H12 and N1-N9 in a SYBR Green rRT-PCRs. H8Nx was identified in three pools collected at the Irene site on 11 February (IRN22-001, IRN22-010, IRN22-013); an H3N1 subtype was identified in a pool collected at Klipdrift dam, Randfontein on 16 February (KD 002), and H4N2 was identified at Bon Accord dam on the 9th of March (BA507). No other subtypes could be determined. ZL204 (17 February) and ZL407 (4 March) from Zoo Lake had lower Ct values of 33.55 and 33.35, respectively, but subtypes H1-H12 and N1-N9 could not be identified here. We did not test for H13-H16 here; alternatively, it is possible that the subtyping primers [29] lack sensitivity for some locally circulating IAVs due to genetic drift.

### 3.2. Phylogenetic and Molecular Clock Analysis

Each of the eight genome segments for the H9N2 viruses from Zoo Lake, namely A/wild bird/South Africa/ZL117/2022, A/wild bird/South Africa/ZL118/2022, A/wild bird/South Africa/ZL121/2022, A/wild bird/South Africa/ZL213/2022 and A/wild bird/South Africa/ZL309/2022 were phylogenetically compared to the available closest reference sequences retrieved from public sequence databases. In the HA gene tree (Figure 2), all five viruses were nearly identical (99.94–100% nucleotide sequence identity) and shared a most recent common ancestor (MRCA) with A/Egyptian goose/South Africa/TP2118/2021 (H9N2), detected at Irene, Pretoria a year earlier on 15 February 2021 [20]. The next closest relative was a recent virus from southern Europe, A/pheasant/Italy/21VIR2284-1/2021 (H9N2). Generally, the Gauteng Province H9N2 viruses from 2021–2022 grouped with H9N1 and H9N2 strains previously detected in healthy wild birds in Asia, Europe, Africa (including South Africa) and Australia. The HA_0_ cleavage site motif was PAVSDRGLF, with the lack of basic amino acids confirming the LPAI pathotype.

The phylogenetic trees for the remaining seven genome segments, carrying the genes encoding PB2, PB1 (and PB1-F2), PA (and PA-X), NP, NA, M (M1 and M2e) and NS (NS1 and NEP) proteins are presented as Appendix A, respectively. In the PB2 gene tree (Appendix A), A/wild duck/South Africa/ZL117/2022 (H9N2) contained proportionally more point mutations compared to the other Zoo Lake viruses, but all five still clustered together with A/Egyptian goose/South Africa/TP2118/2021 (H9N2) as the closest relative. These H9N2 viruses formed part of a larger sub-lineage of viruses exclusively from South Africa, specifically H11N1, H9N2 and H5N2 LPAI viruses detected in commercial ostriches from the Western Cape Province between 2018 and 2020 (sequenced at the University of Pretoria, direct deposit in GISAID). In contrast, the PB1 gene of the Zoo Lake H9N2 viruses sequenced here (Appendix A) had no MRCAs with South African or recent Eurasian viruses (the PB1 gene for A/Egyptian goose/South Africa/TP2118/2021 (H9N2) was not available for comparison). Similarly, there were no MRCAs for the N2 NA gene of the 2021–2022 H9N2 viruses from Gauteng Province (Appendix A). Interestingly, the Gauteng wild bird H9N2 viruses were unrelated to the H9N2 strains detected in the Western Cape Province ostriches in 2019. The latter were more closely related to H1N2, H5N2, H6N2, H9N2 and H10N2 strains from Europe and Asia between 2007 and 2014. The phylogenetic separation of the Gauteng Province H9N2 viruses from others in the PB1 and NA phylogenetic trees most likely indicates gaps in the sequence databases for undetected viruses circulating in the southern African region in recent years.

The PA gene of the H9N2 Zoo Lake viruses (Appendix A) was the only gene sequenced where A/Egyptian goose/South Africa/TP2118/2021 (H9N2) was not the MRCA, indicating that virus genomic reassortment occurred in the Gauteng Province host populations within the past year, but the 2022 H9N2 viruses shared an MRCA with H7N1 LPAI viruses detected in ostriches of the Western Cape in 2020. This South African sub-lineage shared an MRCA with isolate A/mallard/Novosibirsk region/988k/2018 (H4N6), indicating a fairly recent introduction into the South African gene pool of this particular PA gene. The NP (Appendix A) and M (Appendix A) gene phylogenetic trees revealed a South African sub-lineage comprised of the recent H9N2 viruses from 2021–2022 and the H11N1 viruses that circulated in ostriches in the Western Cape Province in 2018–2019, but not other recent ostrich virus subtypes. For NP there was no recent Eurasian virus MRCA for the South African sub-lineage, but for M, A/mallard/Dagestan/1051/2018 (H7N3) was the MRCA, albeit with a bootstrap value of only 55%. For the NS gene (Appendix A), a sub-lineage comprised solely of South African-origin IAVs was again evident, but this time the H9N2 Gauteng wild bird strains shared an MRCA with the H7N1 LPAI viruses from Western Cape ostriches in 2020.

Molecular clock analysis was performed to further investigate the evolutionary relationships between the Zoo Lake H9N2 viruses detected in 2022 and others from South Africa, the African continent and abroad. The tMRCA data from the MCC trees (not shown) was tabulated in Appendix A and combined with the phylogenetic data from Figure 2 and Appendix A to prepare the graphical overview presented in Figure 3. Figure 3 depicts the genetic and temporal relationships between known viruses (black virus diagrams with solid outlines) and hypothetical ancestors inferred by the tMRCA analysis (the nodes) (grey virus diagrams with dotted outlines). The viruses from the Gauteng Province are contained within the dotted black box, with all viruses from South Africa grouped within the dotted green box. Solid black lines connect phylogenetically related genome segments between viruses isolated from/detected in wild birds, represented by green duck figures for South Africa, and lavender duck figures for other countries. Viruses from commercial ostriches are represented by the pink ostrich figures. The node age for the genome segments highlighted in red in the hypothetical ancestral viruses are shown above the virus diagrams (the 95% HPD intervals are given in Appendix A, along with the posterior probability), and the virus names for actual viruses are given underneath the black virus diagrams. Figure 3 is discussed in the next section.

## 4. Discussion

The continuing threat to poultry production and human health of HPAI viruses spread with migrating birds from the northern hemisphere has made wild bird surveillance more important than ever, arguably more so in South Africa where the poultry sector plays a vital role in regional food security. In this study, an environmental faecal sampling strategy was combined with sensitive real-time RT-PCR to survey wild bird populations in the Gauteng Province for IAVs. In Afro-tropical ducks, the IAV detection rate in oropharyngeal vs. cloacal swabs is similar although both rarely simultaneously contain virus [11] but fresh faecal samples collected immediately have similar or higher detection rates than oropharyngeal and cloacal swabs [26]. The use of faecal swabs from the environment has several benefits over traditional capture methods involving baited traps and mist nets required for collecting oropharyngeal and cloacal swabs from live birds. Environmental faecal swabs are not only more cost-effective, simple to collect, flexible, rapid and convenient but also a non-invasive technique that reduces the impact on wild bird habitats and communities and can cover wider geographical areas compared to local-specific sampling sites for live birds [26]. Although we did not apply it in the present study, the disadvantage of not being able to collect information on the bird of origin for samples during environmental faecal sampling can be overcome through the use of mitochondrial DNA barcoding for host bird species identification [26].

The present study was conducted over 18 weeks in the summer and early autumn and overlapped the anticipated peak incidence for IAVs in wild ducks in the province (January/February), and the high-risk period for HPAI virus introduction by Afro-tropical migrant ducks from north of South Africa’s border (February-June). An overall IAV incidence of 42.92% in faecal swab pools was detected, which is substantially higher than other reported IAV incidences in oropharyngeal and cloacal swabs from wild birds in the southern African region during previous surveillance studies. For example, 2.51% in 2007–2009 [21], 5.6% in 2012–2014 [10] and 0.7% in 2018 [18], but our results are consistent with previous surveys using faecal swabs in an Egyptian goose population in Pretoria in 2016/2017 and 2021 [19,20]. The high IAV incidence detected up until May 2022 was unexpected, but anomalous amounts of rainfall in Gauteng Province in April 2022 may have played a role in prolonging virus in wild bird populations during this period. A limitation of our study is that we could not directly compare and determine the trends in monthly prevalence because lower numbers of samples per site were collected in April -May compared to February–March.

The higher percentage of IAV-positive samples we detected compared to prior regional studies [10,11,21] may have been aided by the use of commercial VetMAX AI detection kits. This kit simultaneously targets both the IAV M and NP genes [28] whereas the international method used in the prior surveillance studies in Southern Africa studies only detects the M gene [36]. VetMAX AI detection kits demonstrated superior analytical and diagnostic performance compared to the Spackman test method and others, for example, the limit of viral RNA detection of VetMAX AI detection kits is one to two logs lower [28], and in our experience some of the provincial veterinary laboratories as well as the national reference laboratory that still use the Spackman oligonucleotides consistently fail to detect lower levels of IAV in inter-laboratory comparisons with wild bird samples (unpublished laboratory data). The application of next generation sequencing to swab samples where rRT-PCR Ct values are <30 (i.e., a high viral content) have greatly aided in the identification and genetic characterization of IAVs in wild bird faecal samples and we were able to obtain full genome sequences for an H5N1 HPAI virus and five H9N2 viruses, but a Ct <30 often also results in high rates of successful cultivation from faecal swabs stored in viral transport medium with antibiotics [26,31], and we successfully isolated representative H5N1 HPAI and H9N2 strains that are valuable as antigens for diagnostic testing, vaccine development and other research.

The index case of clade 2.3.4.4B H5N1 was detected on 11 April 2021 in Gauteng Province and within a month the virus had spread via wild birds to other provinces including the southernmost coastline in the Western Cape Province. Outbreaks in poultry and wild birds in South Africa peaked between April and June 2021 but subsided by the end of the year with the arrival of summer. In the first quarter of 2022, three localized outbreaks in the Gauteng Province of H5N1 HPAI occurred in informal/small scale poultry operations on 17 February and 19 April 2022 in the Midvaal Local Municipality (southeast of Johannesburg) and on 15 March 2022 in the Ekurhuleni Local Municipality (east of Johannesburg). H5N1 was also identified in dead wild birds (species unknown) on 15 February in Merafong Local Municipality (west of Johannesburg). Meanwhile, elsewhere in the country the only other reported cases in 2022 were in the Western Cape Province, at one commercial producer in late January, and in coastal seabirds since March 2022 [27]. Our surveillance results confirmed that clade 2.3.4.4B H5N1 HPAI was present in wild birds in the Johannesburg region in late February and March 2022, but also that the virus was present in northern regions of the province in early February 2022, even though no outbreaks in poultry were reported. European White Stork populations appeared to be highly susceptible to the H5N1 HPAI virus, but we did not observe any other sick or dead wild bird species during our sampling. Huge flocks of up to 1200 European White Storks are frequently sighted in the summer months near Johannesburg (eBird, 2022) and at Bon Accord Dam in Pretoria the summer months (pers. obs). European White Storks breed in the Palaearctic region and migrate to sub-Saharan Africa, arriving from October and departing between March and May [37], and this population should be prioritized in future active surveillance.

H9N2 could only be confirmed in bird populations in the southern regions of the province, and the high viral levels detected in early February suggest that the Johannesburg Zoo Lake duck population was immunologically naïve with no prior exposure to the H9N2 subtype. Experimental infections of Egyptian Geese (a species of shelduck indigenous to southern Africa) with LPAI demonstrated that immunologically naïve ducks shed high titres of virus from the respiratory and gastrointestinal tracts for between 7 and 14 days. Ducks that were re-infected with the homologous virus weeks later had acquired subtype-specific adaptive immunity and consequently shed lower amounts of viruses for about 3 weeks. When the ducks were challenged with a heterologous subtype LPAI virus, they had no immunity to the new subtype and high magnitudes of virus were shed from tracheas and cloacae [38]. A closely related H9N2 virus was identified in the Egyptian Goose population Pretoria a year prior therefore it appears that acquired adaptive immunity to H9N2 in the populations around the Pretoria region may have been at played a role in reducing the shed virus levels to below what we could subtype. The H9N2 viruses in Gauteng’s wild bird population in 2021 and 2022 were typical of strains previously described in wild birds across the world and are unrelated to the poultry-adapted H9N2 sub-lineages that have caused disease outbreaks in other parts of Africa, the Middle East and Asia [39].

Figure 3 collates the phylogenetic and tMRCA results for the H9N2 viruses and the related viruses available in the public sequence databases. There is a clear phylogenetic link between the H9N2 virus detected in Egyptian Geese in Pretoria in 2021, the Johannesburg Zoo Lake H9N2 viruses from this study, and H9N2, H11N1, H5N2 LPAI and H7N1 LPAI viruses detected in commercial ostriches in the Western and Eastern Cape Provinces since 2018. Ostriches are farmed extensively in areas concentrated around the Western and Eastern Cape Provinces and are susceptible to infection with IAVs from the wild birds that are attracted into their camps by feed and water, but ostriches are naturally disease-resistant to avian influenza, even HPAI strains [10,14]. Proportionally more IAVs and IAV subtypes have been isolated from ostriches than wild birds in South Africa over the years [10], because commercial ostrich flocks are subjected to compulsory, rigorous and regular IAV testing for international trade purposes, and ostriches therefore act as sentinels for the viruses circulating in the local wild bird population. Figure 3 illustrates that the wild bird IAV gene pool in the distant Gauteng and Western Cape Provinces is genetically linked, which mirrors the documented inter-provincial movements of indigenous water bird species [37]. Although we established that the summer months of January and February are optimal for IAV surveys in the Gauteng Province, the results are not necessarily inferable to other provinces within the country. The Western Cape region for example is a climatic anomaly at the tip of the continent with a winter rainfall climate, in contrast to the rest of southern Africa that has a summer rainfall climate. In southern Africa, most duck species are reproductively active during the late winter and spring in the winter rainfall region (Western Cape) and during the summer in the summer rainfall region (Gauteng and other provinces) [40]. We can therefore speculate that in the Western Cape Province, wild bird IAV surveillance should be focused in the winter months between June and August. This theory is supported by the seasonal trend of peak IAV antibodies detected in ostriches annually in July/August [10], but a longitudinal investigation would be required in the province to confirm it. Other previously described epidemiological features of IAV in South Africa are underscored in Figure 3. Firstly, IAVs are periodically introduced from the northern hemisphere, as evidenced by the MRCAs shared with Eurasian strains. Secondly, gene sub-lineages circulate locally for less than 10 years before becoming extinct [10]. No established regional IAV gene lineages have been detected in this or other studies, with the exception of a chicken-adapted H6N2 sub-lineage has been endemic in South African poultry since the early 2000′s, that to date has not reassorted with wild bird viruses [41]; and finally, South Africa is a true ecological sink for LPAI viruses and reverse gene flow of IAVs from Southern Africa back to Eurasia probably does not occur, since international LPAI virus MRCAs consistently pre-date South African viruses. Nonetheless, it is also evident from the lack of very recent common ancestors for some of the genes that there are significant gaps in the IAV sequence repositories for Africa in general but specifically for sub-Saharan Africa.

In conclusion, active surveillance for IAV is an important element of animal and public health programs, and we demonstrate here that targeted surveillance using environmental faecal sampling from wild water bird populations can be effectively applied under sub-Saharan African conditions. To maximize resources, region-specific studies using sensitive detection technologies should first be used to identify peak prevalence times for IAVs in the local wild bird population. Efforts to detect, sequence and isolate IAVs, not only within South Africa but from other African countries, must be intensified to fill in the gaps and enhance our understanding of IAV’s complex ecology and dynamics in the continent.

## Figures and Tables

**Figure 1 viruses-14-02027-f001:**
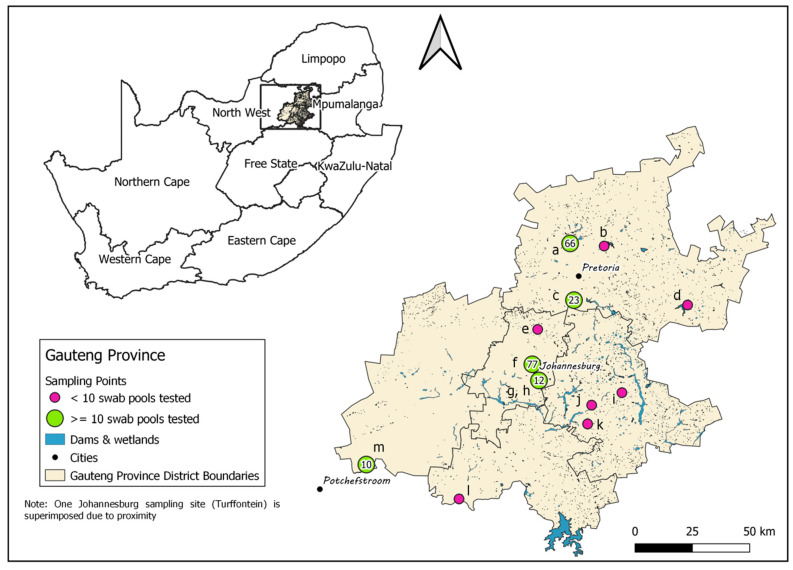
Spatial distribution of thirteen sites investigated for IAV in wild birds in Gauteng Province. The sample size is indicated for those sites where ≥10 swab pools were collected (green points). Locations are indicated as follows: (**a**) Bon Accord Dam, (**b**) Zeekoegat, (**c**) Irene, (**d**) Baja Dam, (**e**) Leeukop Gevangenis Dam, (**f**) Zoo Lake, (**g**,**h**) Turffontein and Wemmerpan, (**i**) Grootvaly, (**j**) Rooikraal, (**k**) Heidelberg Dump, (**l**) Loch Vaal, (**m**) Klipdrift Dam.

**Figure 2 viruses-14-02027-f002:**
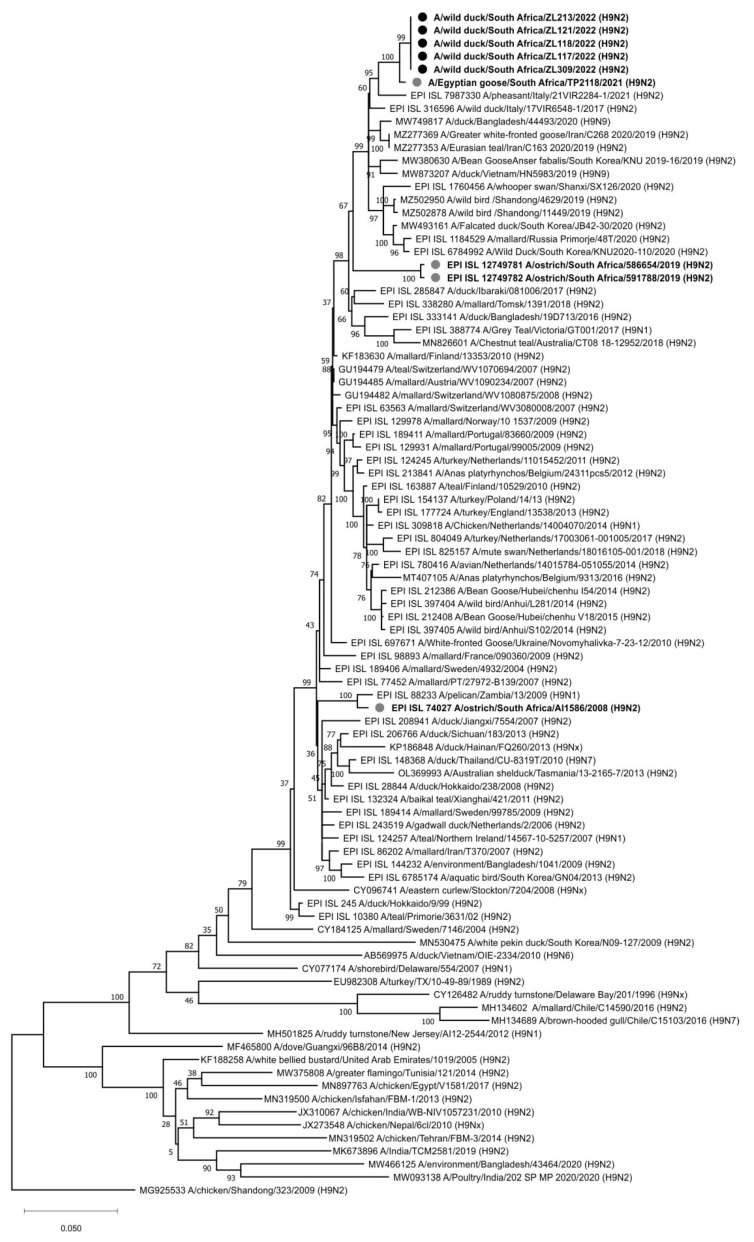
Maximum likelihood phylogenetic tree of the H9 subtype HA genes. Viral sequences from the present study (black dots) and other South African viruses (grey dots) are highlighted.

**Figure 3 viruses-14-02027-f003:**
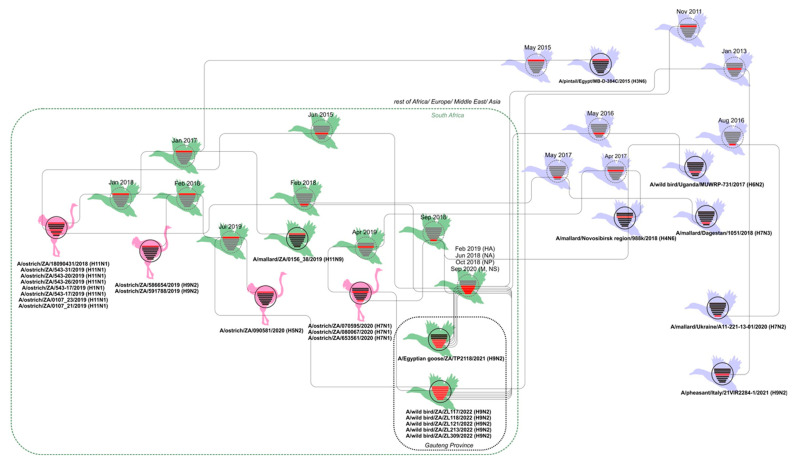
Schematic overview of the ancestry and genomic reassortment between IAV field strains (black, solid lines) and their hypothetical recent common ancestors (grey, dotted lines) interpreted from the ML phylogenetic trees (Figure 2; Appendix A) and tMRCA analyses (Appendix A). The node dates are shown (see Appendix A for the 95% HPD).

## Data Availability

Sequences generated in this study are deposited in the GISAID EpiFlu database under the accession numbers EPI2131759-EPI2131784, EPI2121793-EPI2131801, EPI2131810-EP12131826, EP12131838-EPI2131851 and EPI2131857-EPI2131864.

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
