# Peer review of "Wild Bird Surveillance in the Gauteng Province of South Africa during the High-Risk Period for Highly Pathogenic Avian Influenza Virus Introduction"

_viruses, 2022, doi:10.3390/v14092027_

Round 1

Reviewer 1 Report

       In this manuscript Celia Abolnik et al. reported the surveillance of highly pathogenic avian influenza virus (AIV) in wild birds in the Gauteng province of South Africa between February and May 2022. The authors analyzed 231 faecal swab pools comprising 1,155 individual samples. In this study 100 pooled samples were tested positive for AIVs. The authors detected H5 and H9N2 subtypes as well as H3N1, H4N2, H5N2, H8Nx subtypes. Five complete H9N2 genomes were sequenced, and phylogenetic analyses were conducted.

This study has provided important information on evolution and prevalence of AIV subtypes in the peak rainfall period in the Gauteng province of South Africa. However, there are still a few critical issues that need to be addressed and clarified by the authors.

Major concerns

1.   The authors analyzed the pooled samples from fresh faeces collected from the selected spots in the environment.

       The authors pooled five swabs into 3 ml of viral transport medium. It is known that pooling samples may have a possibility of generating false negative results.  The authors should explain why 5 faeces swabs were pooled instead of more swabs. The authors should discuss the possible impact of sample pooling on the detection outcome, especially the negative results.

2.   In Supplemental Table 1, more than 100 samples were tested negative for the AIV subtypes.

       It is known that RNA integrity may affect the detection of RT-PCR. The authors did not show any data on the quality and quantity of extracted RNA samples. The authors should mention the quality and quantity of RNA samples tested in the RT-PCR assays.

3.    In “Materials and Methods” section, VetMax™-Gold AIV Detection Kit (ThermoFisher Scientific) was used to identify AIV subtypes.

       Based on the data presented in Supplemental Table 1, there are a few samples with Ct value of more than 38 as follows.

              AIV group

D1721-1              Positive (38.47)

BA103         Positive (39.57)

BA104         Positive (39.24)

ZL402          Positive (38.17)

       According to manufacturer’s instruction for the VetMax™-Gold AIV Detection Kit, if a sample’s Ct value is tested > 38 in a real-time RT-PCR, then the sample is considered as “a suspected sample”. Therefore, the sample should be re-tested based on “WorkFlow A or WorkFlow B” to verify whether the sample is truly positive or negative.

I am wondering whether the above samples (Ct value > 38) were re-evaluated and confirmed by a real-time RT-PCR.

Minor concerns

1.   In Results section, 3.1. Detection and identification of IAV in wild bird faecal samples”, please see “…… the complete genome was assembled from Ion Torrent sequencing reads, with deep coverage.”

The “deep coverage” is not an accurate description of sequence reads coverage. The authors should at least mention the average coverage of the sequencing reads in their samples.

2.    In Supplemental Table 1, the authors presented the results as “positive”, “negative” or “not detected”.

       The authors should describe the difference between “negative” and “not detected”.

Author Response

Response to Reviewer 1 Comments

Major concerns

Point 1.   The authors analyzed the pooled samples from fresh faeces collected from the selected spots in the environment.

       The authors pooled five swabs into 3 ml of viral transport medium. It is known that pooling samples may have a possibility of generating false negative results.  The authors should explain why 5 faeces swabs were pooled instead of more swabs. The authors should discuss the possible impact of sample pooling on the detection outcome, especially the negative results.

Response 1: For avian influenza it is not true that pooling samples generates false negative results, and pooling of 5 swabs or more for diagnostics is very common worldwide for cost reasons, and is in fact prescribed in the WOAH Terrestrial Manual. Spackman et al first published work in 2013 on the effects of pooling of chicken swabs, and reported that pooling up to 11 swabs had no significant effect on the detection of influenza virus present. Pieterse et al (2022), cited here, obtained similar results with pooled ostrich swabs. No changes made.

Point 2.   In Supplemental Table 1, more than 100 samples were tested negative for the AIV subtypes.

       It is known that RNA integrity may affect the detection of RT-PCR. The authors did not show any data on the quality and quantity of extracted RNA samples. The authors should mention the quality and quantity of RNA samples tested in the RT-PCR assays.

Response 2: In our view the RNA quality scores would not add any siginificant value to the manuscript because of the nature of the samples from which the total RNA was extracted. Both extraction methods we used are widely-known to deliver good quality RNA that is free of contaminating proteins etc., if the extraction is performed properly. Furthermore, faeces obviously contain a milieu of RNAs derived from the host’s epithelial cells and the countless bacteria, yeast, fungi and viruses present. There is no way to distinguish the proportions of target RNA to contaminating RNA for a “quality” sample here. No changes made.

Point 3.    In “Materials and Methods” section, VetMax™-Gold AIV Detection Kit (ThermoFisher Scientific) was used to identify AIV subtypes.

       Based on the data presented in Supplemental Table 1, there are a few samples with Ct value of more than 38 as follows.

              AIV group

D1721-1       Positive (38.47)

BA103         Positive (39.57)

BA104         Positive (39.24)

ZL402         Positive (38.17)

       According to manufacturer’s instruction for the VetMax™-Gold AIV Detection Kit, if a sample’s Ct value is tested > 38 in a real-time RT-PCR, then the sample is considered as “a suspected sample”. Therefore, the sample should be re-tested based on “WorkFlow A or WorkFlow B” to verify whether the sample is truly positive or negative.

I am wondering whether the above samples (Ct value > 38) were re-evaluated and confirmed by a real-time RT-PCR.

Response 3: No we didn’t retest very low positive samples. In our extensive diagnostic experience these samples would be positive again if the RNA wasn’t stored for too long. As the reviewer points out, only four of the 100 positive fell within the range, and they wouldn’t have a significant impact on the overall positivity if they were truly negative (41.56 % vs 42.92).  No changes made. 

Minor concerns

Point 4.   In Results section, “3.1. Detection and identification of IAV in wild bird faecal samples”, please see “…… the complete genome was assembled from Ion Torrent sequencing reads, with deep coverage.”

The “deep coverage” is not an accurate description of sequence reads coverage. The authors should at least mention the average coverage of the sequencing reads in their samples.

Response 4: The sentence was ammended to “with coverage that ranged from 4472 X to 482684 X per genome segment” to reflect the actual depth of the coverage we obtained.

Point 5 .    In Supplemental Table 1, the authors presented the results as “positive”, “negative” or “not detected”.

       The authors should describe the difference between “negative” and “not detected”.

Response 5: Thank you for piccking up this error in Supplemental Table 1; the two places where “negative” was incorrectly indicated were corrected to “not detected”

Reviewer 2 Report

Dear Editor,

The manuscript entitled "Wild bird surveillance in the Gauteng Province of South Africa during the high-risk period for highly pathogenic avian influenza virus introduction” by  Abolnik et al., reports a study focused on target surveillance of wild birds for avian influenza circulation and introduction. Environmental faecal samples from wild birds were collected in Gauteng Province of South Africa between February and May 2022. Samples were processed with sensitive rRT-PCR methods and positive ones sequenced using NGS approach (Ion Torrent). rRT-PCR methods showed an IAV incidence of 42.92 % (100/231) in the collected and analysed samples; 7 % of the IAV-positive pools tested H5-positive whereas 10 % of the IAV-positive samples were identified as H9N2. Whole genome sequences obtained were used for phylogenetic studies to characterize and survey wild bird populations. 

Some concerns need to be raised, here below some suggestions to improve the quality of the work.

MAJOR COMMENTS

2.4. Virus isolation

Samples with Ct values 30 [26,31] were either inoculated into 9-to-11 day old embryonated Specific Pathogen Free (SPF) hen’s eggs (AviFarms, Pretoria) for virus isolation according to the WOAH-recommended method (2019), or in a confluent QH9-2/1 Quail cells (Nuvonis, Vienna, Austria), according to the supplier’s recommended procedures.: according to “2.2. Sample collection […] Individual droppings were sampled with sterile rayon-tipped swabs […]  pooling five swabs into 13 ml sterile polypropylene tubes (Carl Roth GmbH, Karlsruhe, Germany) pre-filled with 3 mls of viral transport medium (VTM) […].” Nucleic acid extractions and real time RT-PCRs were performed on pooled samples. Here it’s not clear how virus isolation was performed. Did you store and inoculate into 9-to-11 day old SPF hen’s eggs individual dropping samples?

Phylogenetic trees: It is strongly suggested to add outgroup samples; the choice of outgroup sampling is of primary importance in phylogenetic analyses, affecting ingroup relationships and, in placing the root, polarizing characters. Furthermore, I suggest specifying the lineage at which your H9 sequences belongs to (Y439).

MINOR COMMENTS

2.2. Sample collection

[…] pooling five swabs into 13 ml sterile polypropylene tubes (Carl Roth GmbH, Karlsruhe, Germany) pre-filled with 3 mls of viral transport medium (VTM) […]: I suggest to remove “s” character in “mls”.

2.5. Genome sequencing and phylogenetic analysis

[…] Maximum likelihood (ML) phylogenetic trees were generated with the Tamura-Nei model in MEGA-X (v. 10.2.5) [33], with 1000 […]: the number of bootstrap is splitted into page 5 and 6 of the manuscript. Please modify to improve reading.

3.2. Phylogenetic and molecular clock analysis

Each of the eight genome segments for the H9N2 viruses from Zoo Lake, namely A/wild bird/South Africa/ZL117/2022, A/wild bird/South Africa/121/2022, A/wild bird/South Africa/213/2022 and A/wild bird/South Africa/309/2022 were phylogenetically compared to the available closest reference sequences retrieved from public sequence databases. In the HA gene tree (Figure 2), all five viruses were nearly identical (99.94-100 % nucleotide sequence identity): here authors refer to 5 viruses but the names of only four viruses are cited; furthermore “ZL” characters are missing in the names of 121, 213 and 309 viruses.

Author Response

Response to Reviewer 2 Comments

MAJOR COMMENTS

Point 1. Virus isolation

Samples with Ct values ≤ 30 [26,31] were either inoculated into 9-to-11 day old embryonated Specific Pathogen Free (SPF) hen’s eggs (AviFarms, Pretoria) for virus isolation according to the WOAH-recommended method (2019), or in a confluent QH9-2/1 Quail cells (Nuvonis, Vienna, Austria), according to the supplier’s recommended procedures.: according to “2.2. Sample collection […] Individual droppings were sampled with sterile rayon-tipped swabs […]  pooling five swabs into 13 ml sterile polypropylene tubes (Carl Roth GmbH, Karlsruhe, Germany) pre-filled with 3 mls of viral transport medium (VTM) […].” Nucleic acid extractions and real time RT-PCRs were performed on pooled samples. Here it’s not clear how virus isolation was performed. Did you store and inoculate into 9-to-11 day old SPF hen’s eggs individual dropping samples?

Response 1: No we did not collect indididual samples at all, we performed virus isolation on the fluid from the swab pool. To clarify this, on page 5 “Samples” was changed to “Sample pools” to clarify this.

Point 2: Phylogenetic trees: It is strongly suggested to add outgroup samples; the choice of outgroup sampling is of primary importance in phylogenetic analyses, affecting ingroup relationships and, in placing the root, polarizing characters. Furthermore, I suggest specifying the lineage at which your H9 sequences belongs to (Y439).

Response 2: Appropriate outgroups have now been added to each of the ML trees in Figs S1 to S7.

The OIE-FAO OFFLU group is currently revising the classification of H9N2 viruses so that sub-lineage nomenclature can be standardised globally. The first author is a participant in this group, but since the proposed re-classification of H9 hasn’t been finalised yet or officially published, it could not be implemented here (for interest, the Zoo Lake H9N2 viruses are classified as  sub-lineage K.3). We consulted with Dr Isabella Monne at the IZSVe prior to submitting this manuscript to “Viruses”, but she could not recommend a classifcation system to use in the meantime, for this reason no classification beyond “typical wild bird lineage” was used. No changes made.

MINOR COMMENTS

Point 3: 2.2. Sample collection

[…] pooling five swabs into 13 ml sterile polypropylene tubes (Carl Roth GmbH, Karlsruhe, Germany) pre-filled with 3 mls of viral transport medium (VTM) […]: I suggest to remove “s” character in “mls”.

 Response 3: Corrected as suggested

Point 4: 2.5. Genome sequencing and phylogenetic analysis

[…] Maximum likelihood (ML) phylogenetic trees were generated with the Tamura-Nei model in MEGA-X (v. 10.2.5) [33], with 1000 […]: the number of bootstrap is splitted into page 5 and 6 of the manuscript. Please modify to improve reading.

 Response 4: Corrected as suggested

Point 5: 3.2. Phylogenetic and molecular clock analysis

Each of the eight genome segments for the H9N2 viruses from Zoo Lake, namely A/wild bird/South Africa/ZL117/2022, A/wild bird/South Africa/121/2022, A/wild bird/South Africa/213/2022 and A/wild bird/South Africa/309/2022 were phylogenetically compared to the available closest reference sequences retrieved from public sequence databases. In the HA gene tree (Figure 2), all five viruses were nearly identical (99.94-100 % nucleotide sequence identity): here authors refer to 5 viruses but the names of only four viruses are cited; furthermore “ZL” characters are missing in the names of 121, 213 and 309 viruses.

Response 5: Thank you for picking this up. The missing virus (ZL118) was added and “ZL” incorporated where missing.
